# Involvement of Astrocytes in the Formation, Maintenance, and Function of the Blood–Brain Barrier

**DOI:** 10.3390/cells13020150

**Published:** 2024-01-12

**Authors:** Gabriella Schiera, Carlo Maria Di Liegro, Giuseppe Schirò, Gabriele Sorbello, Italia Di Liegro

**Affiliations:** 1Department of Biological, Chemical and Pharmaceutical Sciences and Technologies (Dipartimento di Scienzee Tecnologie Biologiche, Chimiche e Farmaceutiche) (STEBICEF), University of Palermo, 90128 Palermo, Italy; gabriella.schiera@unipa.it (G.S.); carlomaria.diliegro@unipa.it (C.M.D.L.); 2Department of Biomedicine, Neurosciences and Advanced Diagnostics, University of Palermo, 90127 Palermo, Italy; giuseppeschiro1994@gmail.com (G.S.); gabriele96@gmail.com (G.S.); 3Neurology and Multiple Sclerosis Center, Unità Operativa Complessa (UOC), Foundation Institute “G. Giglio”, 90015 Cefalù, Italy

**Keywords:** blood–brain barrier, brain capillary endothelial cell, astrocytes, in vitro BBB models, extracellular vesicles (EVs)

## Abstract

The blood–brain barrier (BBB) is a fundamental structure that protects the composition of the brain by determining which ions, metabolites, and nutrients are allowed to enter the brain from the blood or to leave it towards the circulation. The BBB is structurally composed of a layer of brain capillary endothelial cells (BCECs) bound to each other through tight junctions (TJs). However, its development as well as maintenance and properties are controlled by the other brain cells that contact the BCECs: pericytes, glial cells, and even neurons themselves. Astrocytes seem, in particular, to have a very important role in determining and controlling most properties of the BBB. Here, we will focus on these latter cells, since the comprehension of their roles in brain physiology has been continuously expanding, even including the ability to participate in neurotransmission and in complex functions such as learning and memory. Accordingly, pathological conditions that alter astrocytic functions can alter the BBB’s integrity, thus compromising many brain activities. In this review, we will also refer to different kinds of in vitro BBB models used to study the BBB’s properties, evidencing its modifications under pathological conditions.

## 1. Introduction

The blood–brain barrier (BBB) is a structural and biochemical barrier responsible for the selective passage of molecules from the blood to the brain and for maintaining ion homeostasis in the brain microenvironment [1,2]. As we will discuss below, the central role of the BBB in the transport of metabolites and nutrients from the blood to the brain and vice versa also means that the disruption of its function is involved in most neurological pathologies.

Brain capillary endothelial cells (BCECs), astrocytes, pericytes, microglial cells, and neurons participate in the genesis of the BBB and regulate its properties (Figure 1) [3]. The term neurovascular unit (NVU), which was used for the first time in 2002 [4], refers to the set of all the cellular and molecular components that induce and regulate the formation and maintenance of the BBB [5,6,7,8,9,10,11,12,13,14,15]. In other words, even if the basic structural constituents of the BBB are the BCECs, which form tight junctions with each other and lie on a basal lamina, the other surrounding perivascular brain cells, and in particular astrocytes, play a fundamental role in the formation and maintenance of the BBB, both during brain development and in adult life.

BCECs are held together by two types of junctions: adherent junctions (AJs) and tight junctions (TJs). The AJs perform the function of maintaining cell-to-cell contacts and are also attached to the cytoskeleton. TJs seal the spaces between cells, determine cell membrane polarity, and limit the permeability of the BBB, giving rise to a high transendothelial electrical resistance (TEER) barrier that blocks ions and small charged molecules. And, indeed, measuring TEER using a cell voltmeter is an extremely frequently used method in order to evaluate the formation of a barrier in BBB in vitro models [16].

TJs form a sort of zipper that closes inter-endothelial spaces [1,2]. They consist of integral membrane proteins, such as occludins and claudins, and membrane-associated proteins, such as zonula occludens (ZO-1) [17]. The role of these proteins is of paramount importance for BBB formation [18,19,20,21,22], and, indeed, alterations in their expression or the presence of mutations that compromise their assembly can be pathogenic [23,24].

The transmembrane protein occludin appears to be a major member of the BBB’s TJs, with an important role in the barrier’s function [19,25]. Interestingly, during BBB development, its appearance in TJs is late and, thus, its presence indicates the final maturation of these structures; as a consequence, in BBB models obtained in vitro, its peripheral locations in BCECs are an indication of TJ formation and are frequently accompanied by the appearance of BBB properties in these cultured cells [26,27]. Importantly, TJ components can also bind to other cytoplasmic molecules, and these interactions may contribute towards their assembly [28]. 

BCECs are rich in specific membrane transporters that are involved in the uptake of nutrients, in the removal of waste substances and neurotoxins, and the passage of ions and other molecules. In relation to their different cellular location, there are three main types of transporters and receptors: (a) bidirectional transporters and receptors expressed both apically and basolaterally, for example, glucose transporters (e.g., GLUT-1); (b) unidirectional transporters present in both regions and responsible for the transport of substances either inside or outside the CNS, for example, the insulin or transferrin receptors; and (c) unidirectional transporters located either in the luminal zone or in the ab-luminal zone, which contribute to the polarity of cerebral endothelial cells; the latter category includes the multi-drug-resistant proteins (MDR1s), which are located in the luminal region and prevent the entry of drugs into the CNS by actively expelling them [29].

Moreover, BCECs express a low amount of leukocyte adhesion molecules in order to limit the passage of lymphocytes and other components of the immune system [30].

All these properties appear at different times during brain development and probably require the synergistic effect of all the brain cells. On the other hand, maintaining the integrity of the BBB is of vital importance since its alteration or loss of function underlies various pathologies affecting the CNS, such as neurodegenerative diseases, brain tumors, and stroke. 

In this review, we will consider, in particular, the effects of astrocytes on both the formation and maintenance of the BBB.

## 2. Developmental Appearance and Maintenance of the Barrier’s Function

The exact timing of the appearance of the BBB during development is much debated. One of the most important events that can be associated with the development of cerebral capillaries in rodents is the disappearance of fenestrations and the appearance of TJs in the endothelium between the 11th (E11) and 13th (E13) embryonic days. However, pial vessels show low transendothelial resistance (TEER) up to E20; this suggests that the formation of the BBB is only completed after birth [31]. 

Actually, the formation of the BBB can be considered a two-step process [32]: (i) cerebral angiogenesis is initiated with the entry of the capillaries into the neuroectoderm and the formation of intraneural vessels. This process is defined as “inductive” since BCECs move towards the neuroectoderm, following a vascular endothelial growth factor (VEGF) concentration gradient. VEGF, produced by neuroepithelial cells, binds to the so-called fetal liver kinase receptor 1 (flk-1), also known as VEGF receptor-2, a receptor present on the surface of BCECs; this event directs the BCECs towards the differentiated phenotype (commitment); (ii) consequently, as a result of the interactions with the surrounding cells (e.g., neurons, pericytes, and glial cells), the BCECs will acquire the final phenotype, characterized by the formation of TJs and by the expression of specific molecules. During the development of the cerebral cortex, angiogenesis is followed by vasculogenesis, where vessels originate from pluripotent endothelial cells [33].

In addition to VEGF, essential factors for the genesis and maintenance of the BBB are the Wingless-related integration site (Wnt)/beta-catenin pathway, the G protein-coupled receptor 124 (GPR124), an orphan member of the G protein-coupled receptor family, and the Sonic Hedgehog (SHH) pathway [32].

### 2.1. The General Metabolic Role of Brain Astrocytes

In vivo, astrocytes constitute a sort of protective physiological filter that regulates the flow of metabolites to neurons. Circulating glucose (the main metabolic substrate for the brain) directly reaches the neurons in a very low amount, while it is taken up in a greater percentage by the astrocytes that are in contact with the blood vessels. In other words, astrocytes are ‘intercalated’ between the neurons and glucose, transported into the endothelial cells from the blood by specific transporters (such as GLUT-1). Astrocytes form a large network that embraces many neurons, also at a distance from the BBB; thus, they can transfer glucose to them on a large scale; in addition, in astrocytes, glucose is metabolized by glycolysis into pyruvate and then into lactate, which, once released into the extracellular fluid, can be absorbed by neurons and directly used as an energy source for oxidative metabolism after transforming it back into pyruvate. Interestingly, astrocytes are also the only cells of the nervous system that are capable of storing glucose in the form of glycogen and, therefore, represent an important energy reserve for neurons. Therefore, while the amount of glucose metabolized by astrocytes varies according to the degree of brain activity, the supply of lactate to the neurons is kept almost constant thanks to its continuous release by astrocytes. This process is called astrocyte–neuron lactate shuttle (ANLS), and it seems to be especially active in association with excitatory neurotransmission [34,35,36,37,38,39,40,41,42,43,44]. It has been recently discovered that the consumption of lactate by neurons is essential for long-term memory consolidation but not, probably, for short-term memory [45,46,47,48,49,50]. Interestingly, lactate has also been reported to act as a signaling molecule. A G protein-coupled receptor (GPR81), also named hydroxyl-carboxylic acid 1 (HCA1) or hydroxyl-carboxylic acid receptor 1 (HCAR1), has indeed been discovered and has also been proposed to be involved in processes such as learning, memory, and neuroprotection [51,52,53,54].

Among other functions related to their ability to control metabolite traffic across the BBB, astrocytes have been recently reported to be able to also regulate iron transport into the brain thanks to their production of hepcidin, a peptide that had been considered a liver-specific regulatory factor. This astrocytic ability is extremely important since iron accumulation beyond the amounts required for metabolic activities is considered one of the causes of oxidative stress and, as a consequence, neurodegeneration, while hepcidin is able to control the amount of entering iron, probably by acting, like in the gut, on the ferroportin 1 (FPN1) transporter, present in BCECs [55,56,57,58,59]. As discussed below, they also release growth factors that are able to influence other brain cells and, in particular, BCECs that will form the BBB.

In addition to these important metabolic functions, astrocytes have a fundamental impact on neurotransmission as they are also able to respond to neurotransmitters as well as release their own transmitters (called gliotransmitters) [49,60,61,62,63,64,65]. In other words, a bidirectional exchange of information among astrocytes and neurons also exists at the level of nerve impulses. At a primary level, each synapse is indeed enwrapped by an astrocyte, thus forming what has been called a “tripartite synapse” in which an astrocyte contributes to neurotransmission by not only taking back neurotransmitters, such as glutamate, from the synaptic cleft but probably also by producing and releasing modulatory factors. Moreover, given the particular morphology of astrocytes and their ability to form a web, thanks to the gap junctions (GJs) that bind them to each other, astrocytes can embrace many different synapses and neurons, thus forming a network probably responsible for the many aspects of learning and memory [60,61,62,63,64,65]. More details on this particular aspect of astrocytic function, as well as on the involvement of extracellular vesicles (EVs) in these processes, can be found in a recent review centered on brain cell-to-cell contacts in learning and memory [49].

### 2.2. Astrocytes and BBB Formation and Maintenance

On the basis of the many different observations that suggested an exchange of solutes and water between the interstitial fluid (ISF) and the cerebrospinal fluid (CSF), years ago, the existence of a brain-specific kind of tissue circulation was proposed and called the “Glymphatic System” (GS) [66,67,68,69]. More recently, it is becoming increasingly clear that the GS is essential for maintaining a healthy brain, and, indeed, GS alterations are associated with most neuropathologies [70,71,72].

Now, fundamental for water trafficking across cell membranes are the water channel-forming proteins known as aquaporins (AQPs), but AQPs do not seem to be present in BCECs. On the other hand, AQP 4 is present in astrocytes, in which it is highly polarized, as it is specifically localized at the so-called astrocytic endfeet that contacts the BBB, at the level of very special structures named orthogonal arrays of particles (OAPs) [69]. 

Astrocytic endfeet are very important structures that contain, among other organelles, microtubules, mitochondria, and intermediate filaments composed of glial fibrillary acidic protein (GFAP) [73].

Given its importance for the physiological water trafficking across the BBB, it is crucial that AQP4 proteins are represented at the right levels in astrocytes and that, first of all, they are correctly localized; both alterations in AQP4 expression and its delocalization have indeed been related to pathology. For example, AQP4-deficient mice show significantly higher brain water contents when they are infused with artificial CSF into the brain extracellular space [74]. In addition, it has been reported that the localization of AQP4 also depends on the correct organization of the astrocytic endfeet, which, in turn, depends on the assembly of the gap junctions (GJs), formed by connexins 43 and 30 (Cx43 and Cx30, respectively), between the endfeet; microhemorrhages are indeed more frequent in C43-deficient mice [7]. Actually, brain edema has been evidenced in many pathological conditions, from cancer to stroke, and, in many cases, altered production and/or localization of AQP4 has been found, while, at the same time, the involvement of AQP4 in all these alterations suggests this protein and its organization as a therapeutic target [75,76,77,78,79]. Recently, an AQP4 variant with a C-terminal extension (AQP4x) has been also isolated, and it has been shown that it plays a role in BBB integrity [80]. On the other hand, endothelial cells express the autophagy-related 7 gene (Atg7), an E1-like ubiquitin-activating enzyme that is involved in autophagy [81,82]; Atg7 has been recently found to also regulate the interaction between the astrocytic endfeet and the basal membrane (BM), the extracellular structure through which BCECs and other brain cells communicate: indeed, the dissociation of astrocytes from microvessels in the brain of a transgenic mouse with a conditional deletion of Atg7 in BCECs has been uncovered [83]. Interestingly, Atg7 is involved in the regulation of fibronectin expression, and fibronectin seems to be crucial for the adhesion of astrocytes to the GM [83]. Two other genes, the expression of which is instead astrocytic and, in any case, necessary for astrocyte adhesion to the GM, are those encoding laminin and the laminin receptor [84].

In addition to the mentioned studies concerning the GS and AQPs, many further observations have been suggesting, since a long time ago, a central role of astrocytes in regulating the development and maintenance of the BBB’s functions. For example, in 2008, it was reported that the BBB properties of the brain endothelium in vivo in adult mice depend on signaling mechanisms that involve bone morphogenic proteins (BMPs), which are specifically activated in astrocytes, and the disruption of which causes a loss of the barrier’s function [85]. More recently, it was found that after the tamoxifen-induced apoptotic death of astrocytes in adult mice, BBB damage became evident because of a clear modification of the transport across the BCEC layer [86]. Moreover, in the vessels located close to the apoptotic astrocytes, the expression of TJ proteins (and in particular of ZO-1) was downregulated [86]. It has been recently reported that astrocytes also play a crucial role in BBB maintenance by controlling pH homeostasis through astrocyte-specific proteins, such as the electrogenic sodium-bicarbonate cotransporter 1 (Slc4a4) [87].

Among the astrocytic activities that affect both the developmental formation and maintenance of the BBB, a central role should be attributed to specific growth factors. As we will discuss below, many years ago, by setting an in vitro model of the BBB, we found, for example, that astrocytes (and also neurons) can induce a BBB phenotype in a BCEC cell line when cultured together for a few days [26,27,88,89]. Further studies on this in vitro system allowed us to find out that both cell types produce and release VEGF and fibroblast growth factor 2 (FGF2), which were probably responsible for the effects on the cultured endothelial cells [90,91]. The effects of the astrocyte-derived VEGF as well as of the transforming growth factor beta 1 (TGFβ1) have been further confirmed in recent years [92]. It has been also reported that the astrocyte-derived TGFβ1 can induce, in BCECs, the expression of the TJ protein ZO-1 [93]. Fundamental for BBB maintenance seems also to be the release from astrocytes of Wnt growth factors, which act by stimulating the Wnt/β-catenin pathway, with an important effect on astrocytic endfeet structure too [94]. Another factor recently reported to have a protective effect on the BBB in aged mice after pathological conditions, such as ischemic stroke, is the mesencephalic astrocyte-derived neurotrophic factor (MANF); the authors suggest that the recognition of these MANF functions might offer new ways for approaching ischemic stroke [95]. 

Finally, both astrocytes and their corresponding glial enteric cells are capable of inducing barrier properties in intestinal epithelial cells; S-nitrosoglutathione (GSNO) has been identified as a molecular mediator of this effect [96].

Interestingly, in spite of what has been said above, some aspects of the BBB become functional in vivo before the appearance of the astrocytes during development [97,98]. In this regard, it is interesting to note that the embryonic neural progenitor cells are able to induce BBB properties in BCECs [99]. 

### 2.3. The Role of Extracellular Vesicles (EVs) in the Formation and Maintenance of the BBB

One of the most intriguing discoveries of the last few decades is that all cells, from bacterial cells to the cells of the mammalian nervous system, are able to release extracellular vesicles (EVs) that contain proteins of many kinds and functions, lipids, metabolites, a variety of RNAs, such as microRNA (miRNAs), messenger RNAs (mRNAs), long non-coding RNAs (lncRNAs), even DNA, and sometimes organelles [49,100,101,102,103]; for example, ribosomes have been evidenced in EVs released from oligodendrocytes/Schwann cells and targeted to neuronal axons [104,105,106]. Interestingly, this latter observation has been used to better understand how pre-localized mRNAs, which are a part of RNA–protein complexes, can be translated locally, along the axons and especially at the level of synapses, in response to specific signals [49]. 

Interestingly, in the brain, all the cell types are able to produce and release EVs (Figure 2) that, once received by other cells, can modify their activities and properties both under physiological and pathological conditions [49,107,108,109,110]. 

On the other hand, all the brain cells are also able to catch materials delivered through the EVs under both physiological and pathological conditions [110,111,112,113,114,115,116,117,118,119]. The release of the EVs is not always identical; different signals can indeed modify this process; among other stimuli, it has been, for example, found that hypoxia can induce modifications in the size and total amount of EVs released by both neurons and glial cells as well as in the quality of their cargoes [120]. Similarly, under both central and peripheral stress conditions, many modifications take place in the brain, many of which are spread among cells via EVs [121]. 

Notably, EVs produced by brain cells have been found to also cross the BBB and appear in the peripheral blood; similarly, EVs born in other organs can reach the brain by crossing the BBB [110,122,123]. Although it is not completely clear which are the specific functions of this EV exchange among the brain and the peripheral organs, it is probable that it has a role in the regulation that the brain exerts on all the organs of the body, as well as in the communication to the brain of the status of the different organs [124]. EVs from the periphery have also been found to have beneficial effects on the brain under pathological conditions, as happens, for example, for the EVs released from the mesenchymal stem cells of the bone marrow [124,125,126,127]. On the other hand, EVs that are released from the adipose tissue seem to be responsible for cognitive impairments in diabetes [128].

Even inside the brain, the precise roles of EVs are not completely understood, even if novel activities are continuously discovered or suggested; one of the most important activities of EVs released from both neurons and glial cells (especially from astrocytes) seems to be to modulate synaptic plasticity and hence complex functions such as learning and memory [49,129,130,131]. 

As long as it concerns the formation of the BBB, EVs seem to have a central role; as reported above, years ago, our group found that both neurons and astrocytes produce and release VEGF and FGF2 and that these factors are probably involved in the formation of the BBB in an in vitro system; interestingly, we found these factors in EVs [90,91]. Actually, EVs are released from all the cell types involved in the formation of the NVU, and all these EVs have been suggested to transport molecules important for the formation and maintenance of the BBB, as well as molecules to be eliminated from the brain into the blood [132,133,134,135]. Moreover, the mentioned ability of the EVs to cross the BBB offers the possibility to use circulating EVs as diagnostic biomarkers because of the presence of a few cell-specific surface markers. For example, recently, it has been discussed the possibility that some small non-coding RNAs, present in EVs recognizable as astrocyte-derived (ADEVs), appear to be dysregulated under neurodegeneration-prone conditions [136]. 

Notably, under pathological conditions, EVs may also contribute to altering the BBB and to spread diseases since they can contain toxic proteins, such as amyloid peptides, hyper-phosphorylated Tau proteins, prions, and aggregated α-synuclein [137,138,139,140,141,142,143,144]. In other words, the ability of brain cells to exchange a variety of cell components under physiological conditions is not lost under pathological conditions, but it becomes a sort of weapon that can cause “infection” of neighboring cells.

### 2.4. In Vitro Models Used to Study the Formation of the BBB

In the attempt to clarify the mechanisms involved in the induction of the BBB, various studies have been conducted on both in vivo and in vitro models. It is clear, anyway, that the barrier properties of BCECs are induced during CNS development by the microenvironment in which these cells are placed. Experimental studies have shown that BCECs, if isolated from their natural context and maintained in culture, lose their “barrier” phenotype, probably because of the lack of the epigenetic signals present in vivo and supplied by neurons, astrocytes, and the extracellular matrix (ECM) [145].

#### 2.4.1. Co-Culture Models for Studying BBB Formation and Maintenance In Vitro 

Many studies aimed at understanding the role of the different components of the neurovascular unit were based on in vitro co-culture systems, very often taking advantage of the so-called transwell system (Figure 3). This latter system is based on the use of special culture plates that can also host inserts with a porous membrane at their bottom. Endothelial cells are plated inside the insert, on the porous membrane, enriched with proteins of the basement membrane existing in vivo. Other brain cells can be cultured either at the bottom of the wells (i.e., at a distance from the BCECs) or on the outside of the insert (i.e., very close to the BCECs, even if on the other side of the porous membrane).

Like other groups, by using this transwell system, we set up, in the past, an in vitro model of the BBB, in which the BCECs were initially only co-cultured with neurons [88]. We found that neurons are able to induce a cell line derived from the rat cerebral microvessels (RBE4.B cells) to synthesize and localize occludin to the cellular periphery, and that such a localization is modulated both by the composition of the substrate and by soluble signals released by cortical neurons. These effects do not require close physical contacts between cells; in fact, in the co-culture transwell system used in these experiments, the two cell populations laid at least 1 mm apart [88,89].

In vivo studies identified transport systems for the transfer of amino acids through the endothelial cells co-cultured with neurons; neutral amino acids experience the highest rates of transport [146,147]. On the other hand, the catechol groups provide the molecules with hydrophilicity; for this reason, dopamine and its related catecholamines do not cross the barrier. However, L-3,4-dihydroxyphenylalanine (L-DOPA), the precursor of dopamine, enters the brain from the blood more easily than expected from its fat solubility because of its affinity for the neutral amino acid transporters [148]. It has been reported that both L-DOPA and L-tryptophan permeate the barrier with time-dependent saturable kinetics [149,150,151]. The transfer of L-tryptophan, dopamine, and L-DOPA across the barrier was evaluated by our studies, and it has been seen that the system behaves as a selective interface that excludes dopamine while allowing the passage of L-tryptophan and L-DOPA. Thanks to a mathematical approach, both L-tryptophan and L-DOPA have been shown to move through the BCEC layer through a saturable process, thus indicating that, in the used model, specific carriers are always present. 

Permeation studies have confirmed that the described co-culture system possesses the permeability limits characteristic of the BBB [89]. However, cells must be co-cultured with neurons for at least one week in order to observe a peripheral localization of occludin in endothelial cells [88]. Subsequently, we set up a more complex system with three cell types again based on brain capillary endothelial cells (RBE4.B) and neurons, but also including astrocytes (the system was identical to the one shown in Figure 3c, with astrocytes plated on the outside of the inserts and neurons at the bottom of the plate wells) [26]. These preliminary experiments showed that, in the presence of astrocytes, neuron-induced synthesis and peripheral localization of occludin are earlier (5 days of culture) when compared with the time required when BCECs are cultured with neurons only (7 days of culture). This observation suggested the existence of synergistic effects played by neurons and astrocytes on the barrier’s formation. BCECs grown alone or with neurons and/or astrocytes have shown a different ability to prevent the paracellular passage of sucrose from the donor compartment (i.e., the one to which the sucrose was added) to the acceptor compartment (i.e., the lowest one, in the transwell system, that is separated from the donor compartment by the endothelial cell layer); in particular, it was reported that both astrocytes and neurons are able to independently reduce the paracellular passage of sucrose [27]. Thus, the effects of the two cell types are additive: when BCECs are cultured with astrocytes and neurons, their paracellular flux is reduced by one-third. However, the synergistic effect is only visible after 5 days of co-culture. The quality and/or intensity of the effects of astrocytes and neurons on the BCECs depend on how long the cells have been cultured together [27]. 

The anti-occludin antibodies used in the experiments reported above have a target in an internal region of human occludin (amino acids 132–411); this antibody recognized two different proteins, of which only one (p60) had been expected. The second protein (p48) had never been described before. This protein increases during development even more than p60 and, most interestingly, is enriched in the BCECs cultured with both astrocytes and neurons for 5 days, i.e., under those conditions in which the BCEC layer shows its highest efficiency as a barrier to the passage of sucrose. Although it cannot be excluded that it derives from the degradation of occludin, its abundance, together with the fact that the concentration of p48 changes significantly during development and under different cultural conditions, in relation to the establishment of a functional barrier, suggests that it has a role to play in the formation and maintenance of the BBB [27]. A three-cell-type culture system, including human BCECs, neurons, and astrocytes, and also based on the transwell system, has been more recently described by Barberio et al. [152], who clearly demonstrated the relationship among the presence of astrocytes and neurons and the highest TEER measurements, confirming the existence of a crosstalk between these two cell types and the endothelial cells. Similarly, Ledwig and Reichl reported that three-cell-type models, obtained by using all cell types coming from the same species, also provide a suitable tool for analyzing the permeation properties of new potential brain-targeted drugs [153].

In addition, the interactions between astrocytes and endothelial cells have been studied in a system that also considers the possible effects of a fluid flow (that simulates the blood flow) on BCEC differentiation [154]. In this system, bovine aortic endothelial cells are cultured inside hollow tubes in which the culture medium is allowed to pass, while outside the tubes, there is a chamber that contains the astrocytes (glioma C6 cells). Thanks to this model, it has been shown that this flow plays an essential role in the differentiation of the ECs: it indeed causes an arrest of cell divisions, thus preventing growth on multiple layers; this latter effect is of paramount importance since in vivo the BCECs organize themselves in a single layer. The effect of the flow, added to that of the astrocytes, leads to the reduction in cellular permeability and allows to obtain a system that mimics the real organization of the BBB.

Actually, in vitro co-culture systems allowed for the important demonstration of the effects of different brain cells on the expression of the proteins involved in TJ formation. For example, in a non-human co-culture system in which monkey BCECs were cultured with both rat pericytes and astrocytes, the formation of a barrier was reported based on an increased expression of junctional proteins, such as claudins, ZO-1, and occludin; at the same time, they found an increase in the levels of glucose transporters and ATP-binding cassette-containing (ABC) efflux transporters [155]. The use of an in vitro BBB model also allowed to demonstrate the ability of metformin to cross the BBB, especially under conditions of oxygen–glucose deprivation [156].

Interestingly, co-culture systems prepared from different mammals have shown different levels of TEER; in spite of that, the BBB properties are identical, thus suggesting that different properties, including the species-specific size of BCECs, can be important for determining the ability to control the transendothelial traffic [157].

#### 2.4.2. In Vitro Models Aimed at Studying Modifications of the BBB’s Function under Pathological Conditions as well as the Ability of Drugs to cross a Functional BBB

Recently, in vitro BBB models have been also used to study BBB alterations under pathological conditions, such as cancer, neurodegeneration, or brain injuries [158,159,160,161,162].

As discussed above, the interaction between brain tissue and blood circulation actually takes place on the neurovascular units (NVUs), structures that include different cell types, and their associated extracellular matrix (ECM). In order to create NVU-like structures, pluripotent stem cells (iPSCs) can be induced to differentiate in the NVU cell types and then cultured together in an ECM-like environment [163,164,165]. A similar model, comprising neural and endothelial cells from newborn rats capable of self-assembly in a 3D structure that included a Matrigel ECM, was found to present vascular and BBB-like structures. Strikingly, the obtained NVU could induce vessel formation in the brains of rats suffering from cerebral ischemia [164].

Wevers and colleagues developed a stroke model ‘on-a-chip’ consisting of human BCECs co-cultured with astrocytes and neurons derived from induced pluripotent stem cells (iPSCs). In this model, endothelial cells produce functional transporters and neurons are able to fire. This pathological model was created by inducing hypoxia and hypoglycemia and by blocking the normal fluid flow. Interestingly, under ‘stroke’ conditions, the BCECs increase their permeability and reduce their mitochondrial potential [166]. 

These models have been utilized in the case of ischemic stroke, trying to recreate the cell interactions and the circulatory flow typical of the brain neurovascular units (NVUs) [165]. 

Recently, a new method for the simultaneous isolation of all the NVU cell types from the murine ischemic brain has been described (EPAM-ia method) [167]. Thanks to this method, a NVU transcriptome database was constructed, and it was possible to demonstrate that an osteopontin gene (Spp1) is upregulated after stroke. Interestingly, the increase of osteopontin was also found in stroke patients. Moreover, it was shown that the injection of anti-osteopontin antibodies in the mice reduced brain edema and protected the BBB, opening the possibility that these antibodies could be used in acute ischemic stroke therapy [167]. 

Notably, the use of both mono- and co-culture in vitro systems, as well as the use of media conditioned by brain cells such as astrocytes and pericytes, also suggested that, after an injury, both cell types can release inhibitory molecules that downregulate the production of TJ proteins, thus prolonging BBB dysfunction [168]. 

In general terms, the development of in vitro models of the BBB gives the possibility to study in a controlled environment the relationship among the different components of this very complex system. These in vitro models, and specifically the microfluidic-based ones, are particularly useful for studying pathological conditions because vascular dysfunctions might be directly related to disease progression. This could lead to the possibility of evaluating potentially active drugs and possibly elaborate new therapeutic strategies [169].

In vitro testing platforms, named microphysiological systems (MPSs), have been, in particular, used for the analysis of the ability of drugs to cross the BBB [170,171]. A microfluidic in vitro BBB model (BBB-on-a-chip) has been used to evaluate the possibility to study, in a human in vitro system, the permeability of nanoparticles loaded with therapeutic agents [158,172]. Microfluidic devices have been also used to study the properties of the BBB with intact TJs [173]. In general terms, the use of in vitro transwell systems can be of much help in setting up methods allowing for the transfer of drugs, natural molecules, or even cells across the BBB [11,174]. In one of these systems, the possibility of selecting recombinant adeno-adenoassociated viruses (rAAVs) able to mediate the delivery of recombinant genes to the brain has been successfully studied [175]. In another system, a microfluidic platform has been set up in order to study the ability of lymphocytes to cross the BBB under pathological conditions and how to restore BBB integrity in order to prevent this influx into the brain [176].

In conclusion, we can tell that, in general, the BBB in vitro models could be very useful for not only studying the molecular and cellular mechanisms underlying BBB formation and maintenance but also for studying the capacity of a variety of prodrugs to cross the BBB and acquire a functional structure on the “trans” side of the barrier. 

It is, however, clear that these systems might only partially reproduce the properties of the BBB in a series of contexts. Thus, it is of paramount importance to test in vivo systems in order to confirm the data obtained in the experimental in vitro models. 

## 3. Pathological Alterations of the BBB 

As discussed above, the BBB is a fundamental structure for ensuring and protecting brain functions. However, BBB integrity can be compromised by many pathological conditions, many of which directly arise in the nervous system, while others can be linked to peripheral disorders. Concerning this latter case, it has been found, for example, that chronic kidney disease can cause cognitive impairments by influencing BBB integrity because of an activation of matrix metalloprotease 2 (MMP2) by the high levels of circulating urea [177]. Similar effects have been observed under hepatic encephalopaty, in which hepatic dysfunction affects the integrity of the neurovascular unit [178]. Moreover, hyperglycemic conditions can affect the production of connexin 43 (Cx43), an important component of the gap junctions that bind astrocytes with each other, thus affecting astrocytic properties and, hence, BBB integrity [179]. In addition, it has been found that the depletion of intestinal microbiota, due to antiobitotic use in adult mice, alters the BBB [180]. Notably, some viruses can invade the brain, also causing BBB disruption [181,182,183,184,185,186].

On the other hand, in all the neurological pathologies, from cancer [187,188,189,190] to traumatic brain injury and stroke [191,192,193,194,195], from neurodegeneration (see below) to epilepsy [196], and even in disorders depending on sleep deprivation [197], alterations of the BBB’s integrity have been observed, and in most cases, astrocytes appear to play a central role. As important cellular elements of the neurovascular unit, astrocytes indeed play a prominent role in mediating connections between the endovascular system and neurons and between the immune system and neurons. As a consequence, they also affect brain functioning under pathological conditions, and, in particular, BBB integrity. 

Below, we will first consider the astrocytic alterations observed in Alzheimer’s disease (AD), the most common neurodegenerative disease, and multiple sclerosis (MS), an example of an inflammatory disease. Then, in the third paragraph, we will review some cases of astrocytic alterations under other pathological conditions.

### 3.1. Alzheimer’s Disease and BBB Alteration: A Focus on Astrocytes

AD is the most common neurodegenerative disease in the world, as well as the first cause of dementia; it is mainly characterized by the deposition of extracellular plaques of β-amyloid (Aβ) and by intracellular neurofibrillary tangles (NFTs) of hyperphosphorylated Tau protein. Among other effects, β-amyloid can damage the BBB by aggregating around the vessels, thus also causing glucose transport dysfunction [198]. Moreover, the aggregates of Tau proteins can also alter BBB integrity [199]. In addition, in recent years, other pathological mechanisms have been shown to have a role in the pathophysiology of diseases, such as microvascular disorders, alteration of cerebral blood flow (CBF), and compromised integrity and permeability of the BBB [200,201]. For example, it has been reported that high levels of heparanase in astrocytic endfeet, by inducing an excessive fragmentation of heparan sulfate, can alter the normal process of drainage across the vessel wall [202]. In addition, β-amyloid seems to also be able to stimulate astrocytes to release soluble factors that affect BBB stabilization [203], while inhibiting, on the other hand, the release of FGF-2 [204]. In general terms, altering the ECM components seems to have a role in mediating the damage to both astrocytes and pericytes and, as a consequence, BBB properties in AD [205].

Thus, it seems that BBB dysfunction plays a role in both the onset and progression of AD.

Notably, different structural alterations of astrocytes have been found in AD, probably depending on the microenvironment. Studies on post-mortem brains of familial AD subjects have shown atrophy of astrocytes contributing to the disruption of the BBB, while, in mouse pathology models, around amyloid plaques, astrocytes appear to take on a hypertrophic appearance, with increased cell bodies and processes [206]. The presence of astrocytes around the plaques is probably due to their normal role in phagocytosing amyloid, although in the AD brains, some membrane proteins that internalize amyloid, such as the low-density lipoprotein receptor-related protein 4 (LRP4), are reduced in the reactive astrocytes [207].

Among the mechanisms causing BBB damage, one of the most accredited is metabolic damage, such as, for example, hypercholesterolemia and hypertension [208]. Apolipoprotein E (APOE) isoform 4, a molecule synthesized in the CNS by astrocytes and microglia, and in the periphery by the liver starting from cholesterol precursors, is indeed a risk factor for the onset of AD. It was recently demonstrated, in a specific knockin mouse model, that the expression of APOE4, and not the expression of other isoforms such as APOE2 and APOE3, reduces the astrocyte-mediated protection of the cerebral vessels. The knockout of astrocyte-derived APOE4 restores BBB integrity [209]. Thus, specific protein expression changes in astrocytes can cause an alteration of their functions, affecting, in particular, their ability in maintaining homeostasis of the extracellular environment.

Moreover, in the brain, astrocytes also play a role in controlling neuroinflammation through their interactions with the cells of the immune system. In particular, it has been shown that due to the increased expression of endothelial adhesion molecules, CD4+ and CD8+, T lymphocytes massively infiltrate the areas of the CNS typically affected in AD, and their presence is increased around to sites of Aβ plaque deposition [210]. However, the role of lymphocytes once they have entered the CNS is not yet completely clear. There are, indeed, both reports of deleterious actions and reports of protective functions. It has been shown that CD4+ cells primed with Aβ are able to interact with astrocytes and modify the expression of synaptic proteins in human neuronal-like SH-SY5Y cells, such as by decreasing the expression of synaptophysin. However, a reduction in inflammatory cytokines and a modification of BBB function with changes in claudin and ICAM-1 expression were also observed in astrocytes co-cultured with CD4+ cells [211]. 

### 3.2. Multiple Sclerosis: Astrocytes of the BBB

MS is a chronic inflammatory and degenerative disease of the central nervous system (CNS). Although the exact pathogenetic mechanism of this disease is not yet known, numerous cell types, both resident or not in the CNS, have been shown to be involved at varying degrees in the mechanisms underlying this disease. These cell types can be grouped into three categories: (i) cells of the nervous system, including neurons, oligodendroglia, astroglia, and microglia; (ii) cells of the immune system, in particular CD8+ T lymphocytes, CD4+ T helper-1 and T helper-17 T lymphocytes, and B lymphocytes; and (iii) cells of the vascular system, including the endothelial cells. The crossroad between the immune, vascular, and nervous systems is located at the BBB [212]. Nevertheless, BBB alterations are a hallmark of MS and represent the main pathogenetic mechanism of the relapses. The breakdown of the BBB has a radiological counterpart in lesions evident in magnetic resonance imaging [213]. All the cells that are part of the BBB are involved in the dysregulation of the BBB in MS. Astrocytes, in particular, have shown several pathological involvements in this disease both during the recurrence of relapses and during its clinical progression. In fact, astrocytes interact with cells of the immune system, including self-reactive lymphocytes, and adapt themselves by assuming an inflammatory phenotype [214,215]. The alteration of astrocytes have two main effects, schematically grouped into those resulting from a loss of their homeostatic function and those connected to the acquisition of pathogenic properties. In MS, dysfunctions of the neurovascular unit is also due to the activation of astrocytes that detach from the vessels [216]. In both active and chronic lesions of MS, astrocytes downregulate their production of VEGFA, with a consequent inability to support endothelial growth and stabilization [215]. Moreover, the VEGFA released by astrocytes under the conditions of neuroinflammation could also be deleterious and mediate the leakage of the BBB and endothelial damage [217]. 

It has been shown that pathogenic lymphocytes, particularly Th1 and Th17 lymphocytes, migrate into the CNS and interact with astrocytes at the BBB, suppressing their physiological functions in BBB regulation. In particular, activated astrocytes cause the breakdown of the BBB [215] through the production of different mediators, such as astrocytic thymidine phosphorylase and vascular endothelial growth factor A, that have been shown to disrupt the BBB in the presence of CNS inflammatory lesions [218]. Moreover, reactive astrocytes are also present on the edges of chronically active lesions associated with high clinical disability and increased risk of progression in patients with MS [219]. They have also been found in apparently normal white matter [220], suggesting that astrocytes may infiltrate different areas of the CNS before the onset of damage and are somehow early mediators of the pathology.

Interestingly, EVs released from all the cells of the vessel microenvironment have been reported to have an impact on BBB damage [221], perhaps as carriers of microRNAs (miRs) able to inhibit the expression of mRNAs encoding fundamental factors for BBB maintenance. For example, it has been reported that miR-155 is involved in BBB integrity disruption by inhibiting the synthesis of proteins important for the correct formation of junctional complexes at the BBB [222].

### 3.3. Astrocyte and BBB Alteration under Other Pathological Conditions

As mentioned at the beginning of this paragraph, it has been clearly demonstrated that many pathological conditions, either born in the nervous system or in the periphery, have an effect on BBB integrity. For example, it has been recently found that in methamphetamine (METH)-induced neurotoxicity, neurons not only transfer an excessive amount of aggregated α-synuclein (α-syn) to other neurons but also to astrocytes, where it can induce a decrease in the nuclear receptor-related protein 1 (Nurr1), thus causing an increase in pro-inflammatory factors and BBB damage [223]. Actually, an excess of aggregated α-syn is also normally produced in Parkinson’s disease (PD); thus, also in PD, in which BBB integrity loss is also evident, similar effects on astrocytes and, as a consequence, on the BCEC barrier function might be envisaged.

Similarly, in intracerebral hemorrhage, further damage derives from the formation of perihematomal edema (PHE), which, in turn, causes BBB disruption. It has been found that in PHE, AQP4 expression in the astrocytic endfeet is highly decreased, probably as a response to the increase of reactive oxygen species (ROS) due to the hemorrhage [224]. 

Another pathological condition that involves AQP4 is neuromyelitis optica spectrum disorder (NMOSD), in which patients produce anti-AQP4 antibodies that target astrocytes, thus damaging the BBB, and allowing for further damage to arise due to the chemoattraction of polymorphonuclear leukocytes; notably, however, BBB repair precedes repopulation by astrocytes [225].

Interestingly, it has also been found that when some astrocytes are damaged, the surrounding ones can substitute for them by extending processes that can reach and cover the vascular walls left exposed by lost astrocytes [226]. 

It is also important to underline that senescence with age of the cells that constitute the NVU, and especially of astrocytes, will contribute to BBB damage and, in turn, to neurotoxicity and inflammation [226,227,228,229].

Finally, it is worth noting that it has been reported that ethanol consumption during pregnancy and/or lactation can modify BCEC properties and, in turn, astrocyte gene expression and activities, thus inducing permanent damage to the BBB [230]. Similarly, tobacco smoking and even electronic cigarettes during pregnancy can alter the normal expression of most structural BBB elements, resulting impaired CNS functions, including learning and memory abilities [231].

## 4. Conclusions and Perspectives

In conclusion, we can certainly describe the BBB as a fundamental structure for the maintenance of the environment that allows for all the functions of the central nervous system, from the control of all the body’s physiological activities to more complex abilities such as learning and memory. All the cells that contribute to the microenvironment of the brain capillaries have a function in their structure and function, with many activities, as discussed above, to be attributed to astrocytes. A remarkable observation is that most pathological disorders directly affecting the central nervous system, but also many disorders primarily affecting other organs, can have an impact on the structure and function of the BBB, thus causing, in any case, a worsening of brain activities. Actually, efficient therapies for BBB disruption are not yet available. We suggest that in the near future, an interesting tool might derive from the use of extracellular vesicles (EVs) that might be loaded with a variety of molecules (proteins, metabolites, and drugs, as well as both coding and non-coding RNAs); these EVs might counteract the activities of other molecular species underlying the observed BBB alterations. Importantly, the discussed ability of EVs to cross even an intact BBB in both directions might allow them to treat neurological diseases in very early moments, well before the emergence of pathology-dependent BBB alterations. 

## Figures and Tables

**Figure 1 cells-13-00150-f001:**
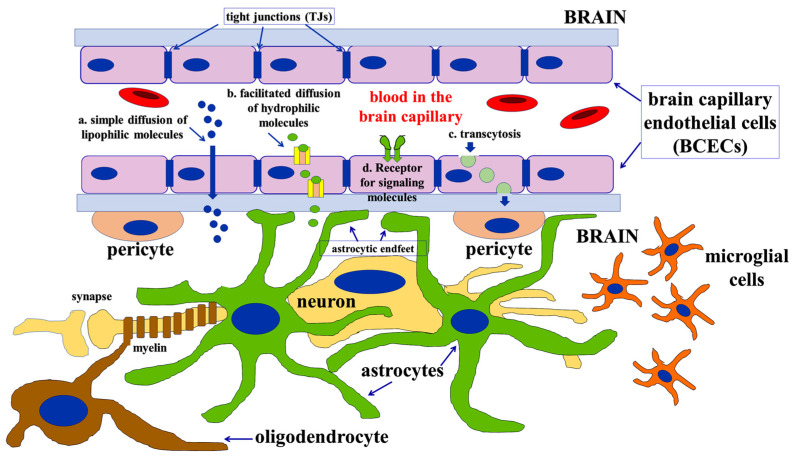
Schematic view of all the components of the neurovascular unit (NVU). The brain capillary is structurally composed of the brain capillary endothelial cells (BCECs) bound together by tight junctions (TJs). The formation of the blood–brain barrier depends, however, on all the cells present around the capillary: neurons (yellow in the picture), oligodendrocytes (brown in the figure), microglial cells (orange in the picture), pericytes (light red in the picture), and, especially, astrocytes (green in the picture). All these cells communicate with each other and with the BCECs by releasing soluble factors that, as discussed below, can be also conveyed by extracellular vesicles. Astrocytes, in particular, directly contact BCECs through the so-called astrocytic endfeet, which also contains aquaporins (see the text for further details).

**Figure 2 cells-13-00150-f002:**
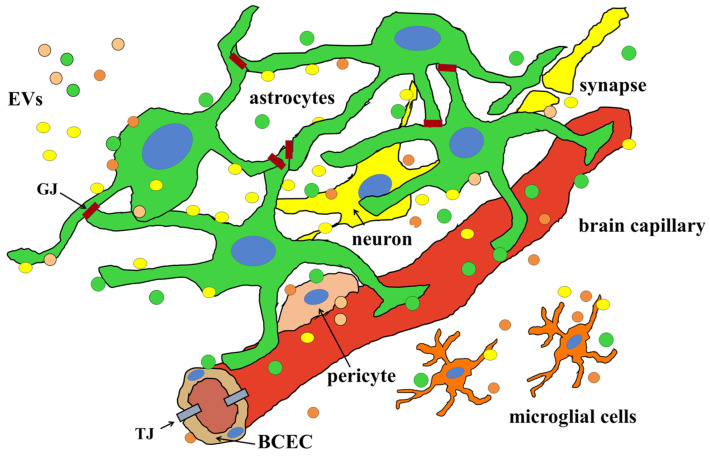
All the components of the neurovascular unit (NVU) are able to release and receive extracellular vesicles (EVs). As discussed in the text, the walls of the brain capillaries are structurally composed of the BCECs bound together by tight junctions (TJs); the formation of a functional blood–brain barrier depends, however, on all the cells present around the capillary. All these cells communicate with each other both through direct contacts and secreted molecules, many of which are delivered by the EVs. For clarity, EVs coming from different cell types have been represented in the same color as the producing cells. Notably, astrocytes form a sort of large web thanks to their ability to form gap junctions (GJs) with each other.

**Figure 3 cells-13-00150-f003:**
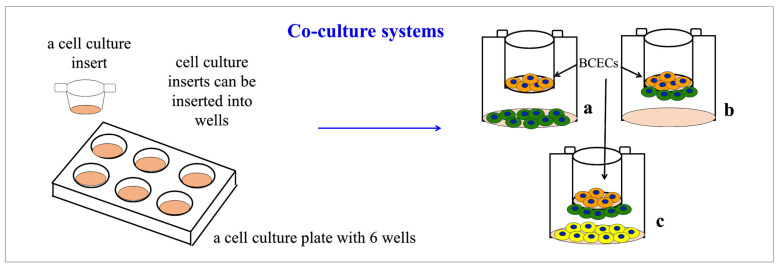
Simple co-culture systems (also called transwell systems) used for studying the role of different brain cell types on BBB formation. This system is based on culture plates with wells that can also host inserts with a porous membrane at their bottom. Endothelial cells (orange in the figure) are plated inside the insert, on the porous membrane, enriched with proteins of the basement membrane existing in vivo. Other brain cells (green in the figure) can be cultured either at the bottom of the wells (**a**) or on the outside of the insert, that is on the other side of the porous membrane (**b**). This system can also be used for studying synergistic effects on the barrier formation of different brain cell types (for example, neurons and astrocytes, or pericytes and astrocytes). These cells can be cultured as a mixed population or separated by culturing one cell type at the bottom of the wells (yellow cells in the figure) and on the outside of the inserts of the other one (green cells in the figure) (**c**).

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
