# Peer review of "Involvement of Astrocytes in the Formation, Maintenance, and Function of the Blood–Brain Barrier"

_cells, 2024, doi:10.3390/cells13020150_

Round 1

Reviewer 1 Report

Comments and Suggestions for Authors

The manuscript provides a comprehensive overview of the critical role of the Blood-Brain Barrier (BBB) in regulating brain composition and function. The authors effectively highlight the structural formation of BBB by brain capillary endothelial cells (BCECs) and emphasize the crucial influence of other brain cells, particularly astrocytes, pericytes, glial cells, and neurons, in the development, maintenance, and properties of the BBB.

However, to enhance the clarity and depth of the review, a few minor revisions are suggested:

  1. Clarification of the specific mechanisms by which astrocytes influence BBB properties and functions would add depth to the discussion. Elaborating on astrocyte-mediated control over neurotransmission and its impact on complex brain functions like learning and memory would strengthen the review.

  2. To provide a more nuanced understanding, discussing recent advancements or specific studies that highlight the ways in which pathological conditions affect astrocyte functions and subsequently alter BBB integrity would be beneficial.

  3. Expanding on the various in vitro BBB models utilized for studying BBB properties and modifications under pathological conditions would add depth to the discussion, possibly by citing specific examples or recent advancements in this field.

  4. Consider addressing the potential limitations or challenges associated with in vitro BBB models to provide a balanced perspective on their utility and relevance in studying BBB dynamics.

To further enhance the comprehensiveness and visual appeal of your review, I would like to suggest the inclusion of illustrative figures or images. Incorporating visual representations such as diagrams, schematic illustrations, or graphs could significantly aid in elucidating the complex interactions between astrocytes and the BBB, as well as in depicting the modifications observed under pathological conditions.

Potential image suggestions could include:

  1. Schematic diagrams illustrating the structural components of the BBB and the interactions between astrocytes and BCECs.
  2. Graphs or charts summarizing the influence of astrocyte functions on various BBB properties.
  3. Representative images or figures depicting in vitro BBB models used for studying BBB properties under pathological conditions.

Including such visual aids would not only enhance the clarity of your manuscript but also provide readers with a more comprehensive understanding of the intricate relationships discussed in the text.

Please consider incorporating relevant images that align with the content discussed in each section to further enrich the manuscript and offer a visual complement to your thorough review.

Overall, these suggested revisions would further enrich the manuscript, offering a more comprehensive understanding of the complex interplay between astrocytes, BBB integrity, and brain function under both normal and pathological conditions.

Author Response

The manuscript provides a comprehensive overview of the critical role of the Blood-Brain Barrier (BBB) in regulating brain composition and function. The authors effectively highlight the structural formation of BBB by brain capillary endothelial cells (BCECs) and emphasize the crucial influence of other brain cells, particularly astrocytes, pericytes, glial cells, and neurons, in the development, maintenance, and properties of the BBB.

However, to enhance the clarity and depth of the review, a few minor revisions are suggested:

  1. Clarification of the specific mechanisms by which astrocytes influence BBB properties and functions would add depth to the discussion. Elaborating on astrocyte-mediated control over neurotransmission and its impact on complex brain functions like learning and memory would strengthen the review.

We added a few sentences underlining the role of the interplay among neurons and astrocytes for neurotransmission and for the processes of learning and memory. Since, learning and memory were not, however, the main aim of the present review, at the end of these additional sentence, we also wrote “More details on this particular aspect of astrocyte functions, as well as on the involvement in these processes of extracellular vesicles (EVs) can be found in a recent review centred on brain cell-to-cell contacts in learning and memory [49]”.

  1. To provide a more nuanced understanding, discussing recent advancements or specific studies that highlight the ways in which pathological conditions affect astrocyte functions and subsequently alter BBB integrity would be beneficial. Expanding on the various in vitro BBB models utilized for studying BBB properties and modifications under pathological conditions would add depth to the discussion, possibly by citing specific examples or recent advancements in this field.

According to this suggestion, we added more examples of in vitro models of pathological conditions in paragraph 2.4 (in vitro BBB models) and more example of astrocyte involvement in pathologies in paragraph 3 (Pathological Alterations of the BBB). Moreover, in order to render clearer our presentation, we subdivided paragraph 2.4 in two different sub-paragraphs, now called, respectively:

2.4.1    Co-culture models for studying BBB formation and maintenance in vitro

2.4.2    In vitro models aimed at studying modification of the BBB function in pathological conditions as well as the ability of drugs to cross a functional BBB

Similarly, we added a third section entitled “Astrocyte and BBB alteration in other pathological conditions” to paragraph 3. New references were obviously added (highlighted in yellow).

  1. Consider addressing the potential limitations or challenges associated with in vitro BBB models to provide a balanced perspective on their utility and relevance in studying BBB dynamics.

As suggested, we added a few comments on the possible limitations of the in vitro models at the end of the corresponding paragraph.

  1. To further enhance the comprehensiveness and visual appeal of your review, I would like to suggest the inclusion of illustrative figures or images. Incorporating visual representations such as diagrams, schematic illustrations, or graphs could significantly aid in elucidating the complex interactions between astrocytes and the BBB, as well as in depicting the modifications observed under pathological conditions. Potential image suggestions could include:

Schematic diagrams illustrating the structural components of the BBB and the interactions between astrocytes and BCECs.

Graphs or charts summarizing the influence of astrocyte functions on various BBB properties.

Representative images or figures depicting in vitro BBB models used for studying BBB properties under pathological conditions.

Including such visual aids would not only enhance the clarity of your manuscript but also provide readers with a more comprehensive understanding of the intricate relationships discussed in the text. Please consider incorporating relevant images that align with the content discussed in each section to further enrich the manuscript and offer a visual complement to your thorough review.

Overall, these suggested revisions would further enrich the manuscript, offering a more comprehensive understanding of the complex interplay between astrocytes, BBB integrity, and brain function under both normal and pathological conditions.

According to this suggestion, we added to new figures:

Figure 1: Schematic view of all the components of the Neurovascular Unit (NVU).

Figure 3: Simple co-culture systems (also called transwell systems) used for studying the role of different brain cell types on BBB formation.

The previous Figure 1 is the new Figure 2.

Reviewer 2 Report

Comments and Suggestions for Authors

This review is about the role of astrocytes in BBB. The BBB contains many components, and its maintenance and regulation includes the role of astrocytes. In particular, the role of astrocytes includes interactions with tight junction and adhesion protein proteins, which are two types of BCEC, and the role of EV (extracellular vesicles).

First, it would be nice to display the overall content in a graph so that reader can better understand the review and/or graphically represent the role of astrocyte in the components of BCEC.

The expression “the authors” appears often. Please indicate in a clearer way (e.g. author's name) or describe it in a different sentence.

It is recommended that the contents of 2.4 (in vitro models used to study formation of the BBB) be divided into chapters.

It is recommended that the impact of astrocytes on the BBB in the diseases mentioned in Chapter 3 be explained in more detail by adding more information related to the BCEC mentioned above.

And, the role of astrocytes in the in vitro BBB model needs to be addressed in more detail.

Comments on the Quality of English Language

There is no major problem with reading English.

Author Response

This review is about the role of astrocytes in BBB. The BBB contains many components, and its maintenance and regulation includes the role of astrocytes. In particular, the role of astrocytes includes interactions with tight junction and adhesion protein proteins, which are two types of BCEC, and the role of EV (extracellular vesicles).

  1. First, it would be nice to display the overall content in a graph so that reader can better understand the review and/or graphically represent the role of astrocyte in the components of BCEC.

According to this suggestion (also done by Reviewer 1), we added to new figures:

Figure 1: Schematic view of all the components of the Neurovascular Unit (NVU).

Figure 3: Simple co-culture systems (also called transwell systems) used for studying the role of different brain cell types on BBB formation.

The previous Figure 1 is the new Figure 2.

  1. The expression “the authors” appears often. Please indicate in a clearer way (e.g. author's name) or describe it in a different sentence.

As suggested, we eliminated the expression “the authors” in almost all the cases: only one remained

  1. It is recommended that the contents of 2.4 (in vitro models used to study formation of the BBB) be divided into chapters.

We subdivided paragraph 2.4 in two different sub-paragraphs, now called, respectively:

2.4.1  Co-culture models for studying BBB formation and maintenance in vitro

2.4.2 In vitro models aimed at studying modification of the BBB function in pathological conditions as well as the ability of drugs to cross a functional BBB

  1. It is recommended that the impact of astrocytes on the BBB in the diseases mentioned in Chapter 3 be explained in more detail by adding more information related to the BCEC mentioned above. And, the role of astrocytes in the in vitro BBB model needs to be addressed in more detail.

According to this suggestion, we added more examples of in vitro models of pathological conditions in paragraph 2.4 (in vitro BBB models) and more example of astrocyte involvement in pathologies in paragraph 3 (Pathological Alterations of the BBB), to which we now added a 3rd section, entitled:  “Astrocyte and BBB alteration in other pathological conditions”. Of course, new references were added to the paper, while adding new examples and discussions.